# In the Eyes of the Beholder—New *Mertk* Knockout Mouse and Re-Evaluation of Phagocytosis versus Anti-Inflammatory Functions of MERTK

**DOI:** 10.3390/ijms25105299

**Published:** 2024-05-13

**Authors:** Sourav Ghosh, Silvia C. Finnemann, Douglas Vollrath, Carla V. Rothlin

**Affiliations:** 1Department of Neurology, School of Medicine, Yale University, New Haven, CT 06520, USA; 2Department of Pharmacology, School of Medicine, Yale University, New Haven, CT 06520, USA; 3Center for Cancer, Genetic Diseases and Gene Regulation, Department of Biological Sciences, Fordham University, Bronx, NY 10458, USA; finnemann@fordham.edu; 4Department of Genetics, Stanford University School of Medicine, Stanford, CA 94305, USA; vollrath@stanford.edu; 5Department of Immunobiology, School of Medicine, Yale University, New Haven, CT 06520, USA

**Keywords:** TAM RTK, MERTK, *Retinitis Pigmentosa*, inflammation

## Abstract

Greg Lemke’s laboratory was one of the pioneers of research into the TAM family of receptor tyrosine kinases (RTKs). Not only was *Tyro3* cloned in his laboratory, but his group also extensively studied mice knocked out for individual or various combinations of the TAM RTKs *Tyro3*, *Axl*, and *Mertk*. Here we primarily focus on one of the paralogs—MERTK. We provide a historical perspective on rodent models of loss of *Mertk* function and their association with retinal degeneration and blindness. We describe later studies employing mouse genetics and the generation of newer knockout models that point out incongruencies with the inference that loss of MERTK-dependent phagocytosis is sufficient for severe, early-onset photoreceptor degeneration in mice. This discussion is meant to raise awareness with regards to the limitations of the original *Mertk* knockout mouse model generated using 129 derived embryonic stem cells and carrying 129 derived alleles and the role of these alleles in modifying *Mertk* knockout phenotypes or even displaying *Mertk*-independent phenotypes. We also suggest molecular approaches that can further Greg Lemke’s scintillating legacy of dissecting the molecular functions of MERTK—a protein that has been described to function in phagocytosis as well as in the negative regulation of inflammation.

## 1. Greg Lemke’s Remarkable Legacy in the Study of TAM RTK Biology

Greg Lemke is one of the pioneers in the field of TAM receptor tyrosine kinase (RTK) biology. TAM is an acronym derived from the first letters of three RTKs—TYRO3, AXL, and MERTK—that are paralogs [1]. The gene duplication events giving rise to three paralogs remain undescribed. Spurred by Cary Lai cloning *Tyro3* [2] in 1991, Greg and his group took an early interest in characterizing this family of RTKs. From what we understand, Martin Gore from the Lemke lab group traveled to Germany and collaborated with Rüdiger Klein to generate *Tyro3*^−/−^ mice. Subsequently, Qingxian Lu in the Lemke group acquired *Axl*^−/−^ and *Mertk*^−/−^ (*Mertk*^tm1Gkm/tm1Gkm^) mice from Stephen P. Goff, Glenn Matsushima, Shelton “Shelley” Earp, and colleagues and, in a tour-de-force, generated *Tyro3*; *Axl*; *Mertk* triple null mice [3]. Decades of superlative research in Greg’s lab on TAM RTKs have defined Greg’s stellar career in science, although his figuratively ‘muscular’ contributions in understanding many other molecules and processes, such as ephrins, evectins, and myelination, are equally impressive [4,5,6,7,8,9]. TAM RTKs have been studied by Greg and colleagues in many areas of biology spanning inflammation and the immune response, viral entry and infection, spermatogenesis and fertility, neurobiology and neuroimmunology, retinal homeostasis and eye diseases, and cardiovascular biology [3,10,11,12,13,14,15,16,17,18,19,20].

## 2. An Introduction to MERTK

The dual roles of TAM RTKs—as regulators of the magnitude of the immune response and mediators of phagocytosis—were apparent during early studies on MERTK. Studies by Camenisch et al. demonstrated increased inflammation in *Mertk* knockout (*Mertk*^tm1Gkm/tm1Gkm^) mice [21]. D’Cruz et al. showed that a strain of rats with defective phagocytosis in the retina harbored a homozygous loss of function mutation in *Mertk* [22] (see below for more on this). Scott et al. subsequently demonstrated loss of apoptotic cell phagocytosis (efferocytosis) in *Mertk*^tm1Gkm/tm1Gkm^ mice [23]. Whether these two roles—as a regulator of inflammation and as a mediator of phagocytosis—are inextricably intertwined or distinct still remains an outstanding question. It is possible that the failure to remove cellular debris or cellular corpses when TAM RTK function is compromised secondarily leads to inflammation. Thus, these functions could be entangled. We (C.V.R. and S.G.), as postdocs in Greg’s and Tony Hunter’s labs, respectively, along with our colleagues, identified a signaling mechanism downstream TAM RTKs that directly inhibits NF-κB signaling and inflammation [11]. Furthermore, Raymond Birge’s group successfully ablated MERTK-driven phagocytosis in cultured cells but observed that there was no increase in NF-κB signaling when phagocytosis was abrogated [24]. These studies imply that the two TAM-dependent functions are likely distinct and indeed separable. The MERTK domain that mediates phagocytosis is the GRB2/PI3K p85/PLCγ-binding domain, but the domain(s) involved in the anti-inflammatory signaling function have yet to be defined [24]. The question of the molecular mechanism employed by TAM RTKs—phagocytosis and/or anti-inflammatory signaling—is relevant to essentially all areas of TAM biology that we have come to know of through the works of Greg and colleagues. For some aspects, it was presumed that TAM RTK function was exclusively in phagocytosis, for example, by retinal pigment epithelial (RPE) cells in continuous turnover of spent photoreceptor outer segment (POS) fragments preventing photoreceptor degeneration [22,25] or by Sertoli cells during spermatogenesis and prevention of infertility [3]. In other contexts, such as the negative regulation of inflammation following T cell activation by activated T cell-derived Protein S (PROS1) acting on TAM RTKs on dendritic cells, it is likely exclusively the anti-inflammatory signaling aspect [26]. Yet another context, such as negative regulation of anti-tumor immunity, may involve more complex MERTK activity affecting both phagocytosis of dead and dying tumor cells and anti-inflammatory signaling. The removal of tumor antigen through efferocytosis by macrophages, for example, as well as anti-inflammatory signaling to create an immunosuppressive tumor microenvironment unfavorable for antigen presentation and unsuitable for T cell activation, may ultimately create an environment that dampens anti-tumor immunity. Altogether, the specification of phagocytosis versus anti-inflammatory functions of TAM RTKs in general and of MERTK specifically remains an important unanswered question relevant to many fields. Here we discuss some recent unexpected findings in the retina and additional evidence from studies in cancer immunity that points to challenges in fully deciphering the molecular functions of TAM RTKs in specific phenotypes and contemplations that are setting the stage to potentially answer such important biochemical and mechanistic questions.

## 3. MERTK in the Daily Phagocytosis of Photoreceptor Outer Segments by Retinal Pigment Epithelia

*Retinitis Pigmentosa* (*RP*) is a heterogeneous group of inherited retinal diseases that affect vision. Vision loss is generally progressive. Symptoms progress from night blindness to tunnel vision and eventually decreased or loss of central vision. One form of autosomal recessive *RP* is caused by homozygous or compound heterozygous mutations in *MERTK* [27,28,29]. This childhood-onset rod–cone dystrophy is now listed as *RP38*. In comparison to most other forms of *RP*, *RP38*, i.e., *RP* due to a MERTK mutation, is characterized by early onset and rapid progression to complete blindness. The molecular understanding of *MERTK* mutations in human *RP* was preceded by observations of a spontaneous animal model now known as the Royal College of Surgeons (RCS) rat. A British company, Bemax Ltd. (Vitamin Foods), located on the Thames in the west London suburb of Hammersmith, produced wheat germ vitamin–protein–mineral supplements from 1927 to the mid-1970s and maintained a rat colony. A particular mutation arose spontaneously in the colony, and the mutant line was brought to Margherita C. Bourne at University College, London, for examination. In a 1938 paper published in the British Journal of Ophthalmology, Bourne and Grüneberg described the degeneration of the retina and cataract as a result of this mutation and characterized the inheritance as recessive [30]. They concluded that “the retinal changes correspond closely to retinitis pigmentosa in man”. Histological assessment and degeneration of the outer nuclear layer in the RCS rat were documented in a 1939 publication by Bourne, Campbell, and Tansley [31]. Both papers emphasized that the development and postnatal maturation of photoreceptors in the retina were indistinguishable from those of wild-type (WT) rats, but that morphological changes began to become apparent around 3 weeks of age.

POS were understood to continuously renew based on the studies conducted by Richard Young [32]. Methionine 3H labeling and autoradiography studies demonstrated that newly synthesized, radiolabeled protein accumulates initially in the inner segment of rod photoreceptors, only to subsequently move as a band to the proximal base of the outer segment and progress through it to disappear at the distal tip of the cell. The close apposition of the POS tips to the RPE and the disappearance of radiolabel from photoreceptors indicated a continuous, routine renewal of the POS and the removal of distal POS tips by the RPE. A schematic of POS engulfment by RPE is shown in Figure 1. Bok and Hall then used electron microscopy (EM) with autoradiography to conclusively demonstrate the phagocytic removal of spent outer segment tips by the RPE and the apparent lack thereof in the RCS rat [33]. Additionally, Herron and colleagues suggested that the defect in the RCS rat is primarily a defect of RPE cells [34]. These investigators used autoradiography to compare rod outer segment growth rates [34]. They observed that the RCS rat POS showed normal growth until about postnatal day (P) 18, following which growth slowed and degeneration set in. Degeneration was progressive and ‘retrograde’, proceeding from the most distal outer segment towards the inner segments. They also reported a lack of indication of phagocytosis by the RPE. Mullen and LaVail conclusively demonstrated a cell-autonomous defect in the RCS RPE [35]. They generated chimeric rats from RCS rats with non-pigmented RPE and WT rats with pigmented RPE [35]. Examination of the resulting mosaic RPE tissue revealed that abnormal or degenerated photoreceptors defined by pyknotic nuclei or outer segment debris accumulated opposite RCS RPE but not normal RPE [35]. More studies on the failure of phagocytosis of the outer segment tips by the RPE followed. Edwards and Szamier fed isolated outer segment fragments from Long Evans (representing WT) or RCS rat retina to primary cultures of Long Evans or RCS RPE cells, followed by EM [36]. While Long Evans RPE phagocytosed outer segment fragments from Long Evans or RCS just fine, RCS RPE failed to engulf outer segment fragments from either. Notably, the same study found no defects in the uptake of polystyrene spheres by RCS RPE cells. Similar results were obtained by Chaitin and Hall using an indirect immunofluorescence approach to differentially label bound versus internalized outer segment fragments fed to Long Evans and RCS RPE cell cultures [37]. Altogether, these results demonstrated that recognition and binding of outer segment fragments are similar between Long Evans and RCS RPE. Still, RCS RPE cells fail to engulf outer segment fragments.

The breakthrough in identifying the gene defect impairing RPE phagocytosis in the RCS rat came in 2000. As mentioned earlier, D’Cruz and colleagues used genetic mapping and positional cloning to identify a homozygous *Mertk* loss of function mutation in RCS rats in March 2000 [22]. Later that year, in December 2000, Nandrot and colleagues reported identical findings [38]. In November of the same year, Gal et al. had already identified three patients with retinal degeneration and mutations in MERTK [27]. Two out of the three mutations were found in homozygosity. The onset of the disease was in early childhood in all three patients. McHenry et al. described another patient with autosomal recessive retinal dystrophy in 2004 and identified compound heterozygous mutations in *MERTK* [39]. Since then, numerous mutations in *MERTK* have been reported to be associated with *RP*.

The role of MERTK in RPE-mediated phagocytosis for retinal homeostasis and the loss of function of this protein resulting in retinal degeneration were consolidated with the recapitulation of the RCS-like phenotype in mice genetically ablated for *Mertk* [40]. *Mertk*^tm1Gkm/tm1Gkm^ mice were generated by Todd Camenisch and colleagues in the Shelley Earp and Glen Matsushima laboratories [21]. These mice were highly sensitive to endotoxic shock [21] and also displayed lupus-like autoimmunity due to a delay in the clearance of apoptotic corpses [41]. Combined with the known phagocytosis defect in *Mertk*^tm1Gkm/tm1Gkm^ mice [23,41], the RCS-like phenotype with rapid retinal degeneration starting in young adult mice [40] was ascribed to the documented deficit in phagocytosis of POS fragments by the RPE in this model.

## 4. In the Eyes of the Beholder? Incongruencies Observed in Some *Mertk* Knockout Mouse Models and Identification of Modifier Alleles for Retinal Degeneration

The studies linking tissue changes, cellular function, and the *Mertk* gene in RCS rats and the similarities thereof in terms of phenotype and loss of function between the RCS rats and *Mertk*^tm1Gkm/tm1Gkm^ mice indicated with certainty that the loss of MERTK alone led to phagocytosis defects in RPE cells. The phagocytosis defect resulted in failure to remove spent POS tips and, consequently, retinal degeneration from rodents to humans. Two discordant studies emerged over the next decade or more. In 2011, Maddox et al. discovered an independent *Mertk* mutant allele (*Mertk*^nmf12^) that did not phenocopy the early onset and rapid progression of retinal degeneration. The mutation was discovered in a screen for retinal degeneration in ENU mutagenized C57BL/6J (B6) mice [42]. The causative mutation mapped to a 3.65 Mb region on chromosome 2 that included *Mertk* (Figure 2). Sequencing the *Mertk* cDNA revealed a missense mutation replacing a highly conserved histidine with arginine in the HRD motif of the tyrosine kinase domain. The failure of the original targeted *Mertk*^tm1Gkm^ allele to complement the retinal degeneration phenotype conclusively demonstrated that the *Mertk*^nmf12^ mutation was the cause of the phenotype. Surprisingly, degeneration observed in *Mertk*^nmf12/nmf12^ mice was limited to the peripheral retina and much less severe than that observed in the *Mertk*^tm1Gkm/tm1Gkm^ line. Electroretinographic (ERG) a- and b-waves were detectable, despite reduced amplitude, and outer nuclear layer (ONL) thickness was preserved in the central retina even in 2-year-old *Mertk*^nmf12/nmf12^ mice, despite the presence of a substantial number of apoptotic (TUNEL-positive) photoreceptor cells by P45. Could this difference, as the authors pointed out, be due to the different genetic backgrounds? The *Mertk*^nmf12^ allele was generated in C57/B6, while the *Mertk*^tm1Gkm/tm1Gkm^ mice were generated from 129 embryonic stem (ES) cells. The authors considered allelic differences as an alternate possibility, with *Mertk*^nmf12^ being a missense mutation as opposed to the presumptive null allele that is *Mertk*^tm1Gkm^. A 2.5-fold reduction in MERTK protein in *Mertk*^nmf12/nmf12^ mice compared to WT B6 mice was demonstrated in the publication. Critical residues essential for RTK activity include the HRD motif [43] (the phenylalanine of the DFG motif makes hydrophobic contacts with the HRD motif) as well as the ATP-coordinating lysine. Given that the mutation in *Mertk*^nmf12^ maps to the HRD motif, this mutation would almost certainly render the *Mertk*^nmf12^-derived protein kinase inactive. Thus, in hindsight, the phenotypic difference was due to differences in genetic backgrounds and not to the missense mutation retaining a significant level of phagocytic function. Work in the Rothlin–Ghosh laboratory later demonstrated that, in fact, kinase-inactive MERTK generated by substituting the ATP-coordinating lysine in B6 mice (*Mertk*^K614M/K614M^) had a similar reduction in macrophage phagocytosis of apoptotic cells as macrophages from *Mertk*^tm1Gkm/tm1Gkm^ mice [44].

In 2015, Vollrath and colleagues demonstrated a clear effect of genetic background as a modifier of retinal degeneration in *Mertk*^tm1Gkm/tm1Gkm^ mice. They noted rare *Mertk*^tm1Gkm/tm1Gkm^ offspring in their colony that had normal-appearing central retinas at 8 months, with degeneration only in the extreme periphery, a similar phenotype to that described by Maddox et al. in *Mertk*^nmf12/nmf12^ mice. ERG confirmed the preservation of both scotopic (rod) and photopic (cone) responses. Vollrath and colleagues hypothesized a single modifier locus genetically linked to *Mertk* based on the segregation of the modified retinal phenotype and the fact that it had persisted in the colony despite six generations or more of backcrossing to B6. Indeed, by using short tandem repeat (STR) markers, they uncovered a difference in the chromosomal region surrounding the *Mertk*^tm1Gkm^ allele that correlated with the retinal phenotype (Figure 2). They found that *Mertk*^tm1Gkm/tm1Gkm^ mice with pan-retinal degeneration as in the original line [40] were homozygous for 129 alleles across a 40 cM segment of chromosome 2 segment (*D2Mit94* to *D2Mit168*), consistent with the 129 ES cell origin of the knockout allele. In *Mertk*^tm1Gkm/tm1Gkm^ mice with normal-appearing central retinas, a large segment of this region (*D2Mit94* to *D2Mit445*) was now homozygous for B6 alleles, likely due to meiotic recombination in a *Mertk*^tm1Gkm/+^ mouse during backcrossing. This suggested that the modifier was located between *D2Mit94* and *D2Mit445* and that homozygosity for the B6 allele of the modifier suppressed retinal degeneration except in the extreme periphery. Interestingly, *Mertk*^tm1Gkm/tm1Gkm^ mice heterozygous (129/B6) for the modifier locus showed an unusual intermediate phenotype; intermixing in the non-periphery of one to four islands of degeneration surrounded by normal-appearing regions was found in 17 of 20 mice examined [45]. Normal-appearing regions in these mice contained greater than 2-fold more phagosomes in the RPE than degenerating regions, consistent with restoration of phagocytosis by the modifier. In summary, these studies identify that severe, early-onset retinal degeneration and vision loss upon loss of *Mertk* in mice were dependent on genetic background and not solely due to the absence of the *Mertk* gene.

Vollrath and colleagues used meiotic backcross mapping to narrow the location of the modifier to an approximately two megabase critical interval between *D2Mit445* and *D2Mit62*. Importantly, one of the >50 genes in this interval is *Tyro3*. Consistent with *Tyro3* as a modifier gene, the retinal degeneration phenotype was more severe when *Mertk*^tm1Gkm/tm1Gkm^ was crossed with *Tyro3*^−/−^ (targeted knockout) mice to generate *Mertk*^tm1Gkm/tm1Gkm^; *Tyro3*^−/−^ than in *Mertk*^tm1Gkm/tm1Gkm^; *Tyro3*^129/129^ mice. Conclusive evidence that *Tyro3* is necessary for the modifier effect came from examining *Mertk*^tm1Gkm/tm1Gkm^; *Tyro3*^129/B6^ versus *Mertk*^tm1Gkm/tm1Gkm^; *Tyro3*^−/B6^ offspring from a *Mertk*^tm1Gkm/tm1Gkm^; *Tyro3*^B6/B6^ x *Mertk*^tm1Gkm/tm1Gkm^; *Tyro3*^−/129^ cross, which demonstrated a more severe retinal degeneration phenotype in *Mertk*^tm1Gkm/tm1Gkm^; *Tyro3*^−/B6^ mice. Interestingly, females are more severely affected than males in both classes. The authors went on to demonstrate that the RPE of *Tyro3*^129/129^ mice had about one-third the amount of TYRO3 protein of *Tyro3*^B6/B6^ mice, suggesting a model in which increased TYRO3 protein from the B6 allele suppressed retinal degeneration in all but the periphery of *Mertk*^tm1Gkm/tm1Gkm^ mice. The model explains the suppressed central retinal degeneration in *Mertk*^nmf12/nmf12^ mice, which are on a B6 background and therefore *Tyro3*^B6/B6^. Consistent with this, immunofluorescence demonstrated decreased TYRO3 signal in the central region of *Mertk*^tm1Gkm/tm1Gkm^; *Tyro3*^129/129^ mice compared to *Mertk*^tm1Gkm/tm1Gkm^; *Tyro3*^B6/B6^, whereas peripheral staining was similarly diminished for both genotypes. Differential expression of genes between the macula and the periphery is known to occur in human eyes, although *MERTK* expression was found to be invariant [47]. TYRO3 expression was not investigated in the study [47]. It is interesting to note that the Lemke lab also reported ~3× reduction in *Tyro3* mRNA level and an absence of TYRO3 immunostaining in *Mertk*^tm1Gkm/tm1Gkm^ RPE [18], albeit the hypomorphic expression of *Tyro3* from the 129 allele remained unknown to them.

*Tyro3*, being a paralog of *Mertk*, was the obvious suspect. In theory, another gene located in the critical interval could also be presumed guilty. In vitro, ectopic expression of murine TYRO3 drove rat NRK-49F fibroblasts to phagocytose approximately twice as many bovine POS fragments relative to a GFP control [45]. Expression of TYRO3 in primary cultured RPE from *Mertk*^tm1Gkm/tm1Gkm^; *Tyro3*^−/−^ mice also increased POS phagocytosis by approximately 2-fold, although the efficiency of phagocytosis in similar experiments with expression of MERTK was somewhat higher [45].

## 5. Beyond the Eye: The Role of MERTK in Anti-Tumor Immunity

Rothlin, Ghosh, and colleagues have also been studying *Mertk* and TAM RTKs. Their focus has primarily been on the immunological roles of these RTKs. They and others developed the concept that TAM RTKs, including *Mertk*, constitute an innate immune checkpoint [48,49,50,51,52,53,54,55]. CTLA-4 and PD-1 function as immune checkpoints by driving T cell exhaustion, so they proposed that TAM RTKs function in innate immune cells to rein in inflammation and thereby the magnitude of the immune response. Inhibition or blockade of CTLA-4 and PD-1 checkpoints dramatically impacts cancer therapy. The Rothlin–Ghosh group had hoped that inhibition or blockade of TAM RTKs would similarly benefit cancer therapeutics. Initial results using the *Mertk*^tm1Gkm/tm1Gkm^ mice were astonishing. Reports demonstrating remarkable anti-tumor resistance in *Mertk*^tm1Gkm/tm1Gkm^ mice started to be published [48,49,50,56,57,58,59,60,61,62]. The Rothlin–Ghosh group tried several subcutaneously implanted cancer models, including the CTLA-4 and PD-1 checkpoint inhibitor-resistant YUMM1.7 and an orthotopic glioblastoma model in these mice [46]. These models revealed a remarkable anti-tumor immune response that curtailed tumor growth [46]. This initial excitement dissipated with the generation of a *Mertk*^fl/fl^ mouse and attempts to probe the cellular basis of MERTK function in anti-tumor immunity. Of note, the *Mertk*^fl/fl^ mouse line was generated by targeting B6 ES cells, the technology having progressed sufficiently to do so at that point. Despite valiant efforts by a graduate student, Yemsratch Akalu, to ablate *Mertk* in dendritic cells, myeloid cells, and endothelial cells, the proposed anti-tumor immunity observed in *Mertk*^tm1Gkm/tm1Gkm^ mice was not recapitulated. Of course, MERTK function may be required in more than one cell type for anti-tumor immunity. But with the known issues of generating mouse knockouts with 129 ES cells and the results of Maddox et al. and Vollrath et al., the Rothlin–Ghosh group decided to generate a germline ablation with the B6 ES cell-derived *Mertk*^fl/fl^ mice (*Mertk*^−/−V2^; Figure 2). Once germline knockouts were obtained and Yemsratch repeated the tumor implantation experiments, the anti-tumor immunity was not observed again [46]. Using another independently developed *Mertk* knockout mouse using B6 ES cells (*Mertk*^−/−V3^), the authors concluded that the anti-tumor immunity against YUMM1.7 and GL261 that they noted was ***not*** due to the loss of MERTK [46].

## 6. Retinal Degeneration Exclusively Due to the Loss of *Mertk* Re-Examined

These mouse lines were also used to re-investigate the retinal phenotype [46]. Histology, EM, and ERG studies demonstrated that the new *Mertk*^−/−V2^ and *Mertk*^−/−V3^ mouse lines generated using B6 ES cells did not display the severe, early-onset retinal degeneration characteristic of the 129 ES cell-derived *Mertk*^tm1Gkm/tm1Gkm^ mice [46]. The 129 ES cell-derived *Mertk*^tm1Gkm/tm1Gkm^ mouse line in the Rothlin–Ghosh lab had a 129 region spanning *D2Mit206* to *D2Mit168* [46]. Bulk RNA sequencing for whole genome transcriptomics demonstrated significant expression differences between 129 ES cell-derived *Mertk*^tm1Gkm/tm1Gkm^ mice and B6 ES cell-derived *Mertk*^−/−V2^ and *Mertk*^−/−V3^ mice for genes expressed on chromosome 2, as well as possibly in chromosomes 18 and 19 [46]. Moreover, these gene expression differences were also tissue-specific, differing between RPE and bone marrow-derived macrophages (BMDMs). The basis for changes in the expression of genes in chromosomes 18 and 19 remains unknown. Finally, CRISPR-targeting to disrupt *Tyro3* in *Mertk*^−/−V2^ B6 ES cells and the generation of fully B6 *Mertk*^−/−V2^; *Tyro3*^−/−V2^ mice (Figure 2) unequivocally supported the notion that the retinal degeneration observed in 129 ES cell-derived *Mertk*^tm1Gkm/tm1Gkm^ mice required the simultaneous loss of function of MERTK and diminished expression of TYRO3 [46].

## 7. Inflammation in the *Mertk*^−/−^ Retina

Inflammation was suspected to occur early postnatally in the retina of RCS rats and was examined by the Finnemann lab. Indeed, the RCS rat retina and the *Mertk*^tm1Gkm/tm1Gkm^ retina were found to exhibit elevated levels of pro-inflammatory cytokines such as CCL4 and CCL5 as early as P14, an age prior to the onset of routine outer segment renewal. Tamoxifen administration has been shown to inhibit retinal microglial activation, slowing retinal degeneration in mice with acute or inherited retinal degeneration [63]. In agreement, tamoxifen eyedrops and/or targeted toxic liposomes were effective in reducing microglia activation in the RCS rat retina. Such treatment delayed retinal degeneration in the RCS rats by prolonging photoreceptor survival and function, as established by ERG recordings. These findings point to the involvement of microglia and, consequently, inflammation in the process of RCS rat retinal degeneration [64]. Signs of retinal inflammation were apparent in the RCS rat retina before distortion of POS, let alone accumulation of POS debris. Whether inflammation was secondary to the phagocytic failure of the RPE remains uncertain. Chronic and late-stage inflammation have been suggested in the pathology of various forms of *RP* [65]. Indeed, *Retinitis* refers to inflammation of the retina. Nonetheless, Mercau et al. observed increased inflammatory gene expression within the RPE of *Mertk*^tm1Gkm/tm1Gkm^ mice at P10, even before eye opening and when outer segments are not yet mature [66]. This result shows an early inflammatory event starting within the RPE. 

The study by Mercau et al. found that, in mice, only the simultaneous loss of MERTK and TYRO3 leads to RPE inflammation and retinal degeneration [66]. The inflammatory response was found to evolve from the early increased expression of genes involved in the response to Type I Interferons, TNF-α, and IL-6 in the RPE at P10 to a broader response characterized by an increase in the number of microglia, with their activation and translocation to the ONL by P25. This inflammatory response preceded any measurable changes in the thickness of the ONL. Once photoreceptors died and the ONL thickness was significantly reduced, a more expansive inflammatory response, including monocyte infiltration in the retina, was detected. Exacerbated inflammation was observed when both MERTK and TYRO3 were lost, but not when either MERTK alone or TYRO3 alone was lost. Of note, *Tyro3*^−/−^ mice were also 129 ES cell-derived. Enhanced inflammation only in the absence of both MERTK and TYRO3 implies functional redundancy between these two paralogs. This redundancy may occur at any time. Alternatively, even though no increase in TYRO3 protein is noted in *Mertk*^−/−V2^ or *Mertk*^−/−V3^ mice, TYRO3 activity may take over the function once MERTK is lost. The later interpretation is likely since the retina is normal in *Tyro3*^−/−^ mice [18,66]. Given the structure and function of these RTKs, the redundancy/compensation is perhaps not surprising. What is less so is that Mercau et al. observed a dramatic defect in the phagocytic activity of the RPE in their B6 ES cell-derived *Mertk*^−/−V2^ mice when compared to WT B6 mice at the time of the daily burst of phagocytosis after light onset. The complex cascade that ensues when phagocytosis is affected, such as in *Mertk*^tm1Gkm/tm1Gkm^ mice or *Mertk*^−/−V2^; *Tyro3*^−/−V2^ mice, makes comparison with B6 ES cell-derived *Mertk*^−/−V2^ mice not feasible, hence the comparison of RPE phagocytosis was only between B6 WT and B6 ES cell-derived *Mertk*^−/−V2^ mice. These results suggest that TYRO3 functions in attenuating inflammation but not in promoting phagocytosis. This is inconsistent with increased phagosomes observed in normal-appearing retinal regions of *Mertk*^tm1Gkm/tm1Gkm^; *Tyro3*^129/B6^ mice and with results from in vitro expression of TYRO3 in cultured human RPE cells that promoted outer segment phagocytosis [45]. Given the structural similarities of TYRO3 protein with MERTK, including the GRB2/PI3K p85/PLCγ-binding domain, it is unclear why TYRO3 would not function in phagocytosis, especially since TYRO3 receptors colocalize with ezrin at the apical microvilli of mouse RPE cells [18]. The in vivo functional specification of TYRO3 needs to be investigated in more detail.

Inflammation in the RPE was observed as early as P10 [66]. This is a time point that precedes eye-opening in rodents. This very early onset of inflammation in the RPE suggests that RPE cells themselves are the first responders to inflammatory cues. Although microglia are activated in *Mertk*^tm1Gkm/tm1Gkm^ and *Mertk*^−/−V2^; *Tyro3*^−/−V2^ mice, this is observed later than RPE inflammation [66]. Furthermore, genetic ablation of *Mertk* in microglia and myeloid cells in mice (*Csf1r*-cre^+^; *Mertk*^fl/fl^ mice) did not result in inflammation or retinal degeneration [66]. Thus, microglial activation is likely a consequence of RPE inflammation in *Mertk*^tm1Gkm/tm1Gkm^ and *Mertk*^−/−V2^; *Tyro3*^−/−V2^ mice. In microglia, *Tyro3* is not expressed. Therefore, the effects of ablation of *Tyro3* in these cells were not tested. In conclusion, results published to date support the current hypothesis that inflammation in the absence of MERTK and TYRO3 starts in the RPE. It then spreads to microglia and eventually results in the recruitment of monocytes. An inexorable cascade of uncontrolled inflammation, perhaps in a feedback manner, ultimately severely damages the retina.

Notwithstanding the clear evidence for RPE inflammation in the absence of MERTK and TYRO3 at P10, the driver of this inflammation is not understood. MERTK and TYRO3 can dampen inflammation, but this activity is only important when there is a driver of inflammation. There are a number of inherent or pathophysiological processes involving the RPE that yield retinal inflammation, but none of the known culprits are present as early as P10. An obvious candidate is oxidative stress, with reactive oxygen species (ROS) generated due to the light reactivity of phototoxic compounds or even as a by-product of mitochondrial electron transport [67]. However, ROS are generally believed to be damaging if continuously elevated and important for aging or severe pathologies such as acute light damage. There is no evidence of elevated ROS in the RPE at P10. Another candidate may be defective processing of visual pigments and loss of photoreceptor homeostasis, which in turn drives inflammation. The RPE converts all-*trans*-retinol into retinyl esters, then into 11-*cis*-retinol, and finally into 11-*cis*-retinal. Low levels of 11-*cis*-retinal and high levels of all-*trans*-retinyl esters have been reported in *Mertk*^tm1Gkm*/tm1Gkm*^ RPE, as compared to WT, by 3 months of age [68]. By this age, significant numbers of photoreceptors have already degenerated. Autofluorescence attributed to retinyl esters at the interface of RPE and photoreceptor cells at 6 weeks of age in albino *Mertk*^tm1Gkm*/tm1Gkm*^; *Rpe65*^Met450^ and *Mertk*^tm1Gkm*/tm1Gkm*^; *Rpe65*^Leu450^ mice, but not in agouti *Mertk*^tm1Gkm*/tm1Gkm*^; *Rpe65*^Met450^ mice, was also described in an independent study [69] that implicated toxic bisretinoids in the degenerating retina of *Mertk*^tm1Gkm*/tm1Gkm*^ mice. RPE cells may generate bisretinoids such as di-retinoid-pyridinium-ethanolamine (A2E) from all *trans*-retinal and phosphatidylethanolamine. RPE-resident A2E and its derivatives can be a source of inflammation. RPE A2E was notably higher in *Mertk*^tm1Gkm*/tm1Gkm*^ eyes at 2 months of age when compared to WT [68]. In the absence of MERTK, a defect in outer segment phagocytosis may lead to increased 11-*cis*-retinal or all-*trans*-retinal within photoreceptors and thus more A2E uptake by the RPE prior to photoreceptor loss. Incidentally, A2E uptake is MERTK-independent [68]. This is consistent with elevated levels of A2-glycero-phosphoethanolamine (A2GPE) and A2E in 6-week-old albino *Mertk*^tm1Gkm*/tm1Gkm*^; *Rpe65*^450Leu^ eyecups [69]. The absence of anti-inflammatory signaling may fail to constrain inflammation due to A2E accumulation in the *Mertk*^tm1Gkm*/tm1Gkm*^ retina. Neither visual cycle defects nor A2E formation are likely to play a role at P10. A final molecular mechanism associated with the RPE and with retinal inflammation that we wish to highlight is the intracellular processing of phagocytosed POS through LC3-associated phagocytosis (LAP) [70,71]. Following integrin/MERTK-dependent engulfment, internalized POS phagosomes in the RPE share marker proteins typically associated with autophagosomes such as *Atg5*, and such proteins are important for efficient POS degradation and the activity of the visual cycle. LAP deficiency in macrophages is pro-inflammatory [72]. Moreover, autophagy can inhibit NLRP3 inflammasomes [73,74], and NLRP3 can be activated by lipid droplets, such as retinosomes [75,76]. RPE-specific ablation of *Atg5* resulted in ERG deficits only by 16 weeks of age but not at 5 weeks of age, suggesting the progressive effects of continuous disruption of diurnal POS turnover. Of note, MERTK has been directly implicated in the inhibition of NLRP3 inflammasome activation by upregulating autophagy [77]. Again, such defects are unlikely to account for the inflammation detected at P10. Altogether, RPE insults associated with elevated oxidative stress, abnormalities in the visual cycle, or phagolysosomal processing cause retinal inflammation, and their effects may be moderated by MERTK signaling. Nevertheless, the cause(s) of the RPE-specific inflammation found at P10 in mice lacking both TYRO3 and MERTK are unrelated to these known insults and remain to be identified. The discovery of inflammation as a critical aspect of retinal degeneration associated with the loss of MERTK might spur the investigation of anti-inflammatory therapies for alleviating vision loss. Jakinibs [66] or autophagy inhibitors [77] can be used to suppress inflammation resulting from the loss of MERTK.

## 8. How Can MERTK Function in Phagocytosis versus Downregulation of Inflammation Be Dissected?

The dual roles of MERTK as an efferocytosis receptor and as an anti-inflammatory signaling molecule, as already touched upon earlier in this review, leave an outstanding question. How can these functions be dissected to find out which of these dual roles accounts for a specific biology associated with MERTK? The observation of inflammation possibly being a core component along with failed phagocytosis in the absence of TAM RTK, such as in the RPE, highlights this need. Only such an approach can distinguish the primary effects from the secondary effects. For example, does loss of phagocytosis lead to inflammation? Of note, phagocytosis is not inevitably anti-inflammatory. For example, the phagocytosis of infected apoptotic cells is pro-inflammatory [78,79,80]. Can inflammation result in the loss of phagocytosis? We believe that the best approach will be to use mouse models with point mutations that specifically disable one function, e.g., phagocytosis, while leaving the other function intact. We have reported on the generation of a MERTK point mutant that targets its kinase activity using B6 ES cells [44]. This mutation disrupts phagocytosis by BMDMs to the same extent as seen in germline *Mertk*^−/−^ mice (B6 ES cell-derived). Of note, TYRO3 is not expressed in BMDMs, and MERTK is responsible for ~50% of the efferocytosis activity of these cells. This mutant can be anticipated to also have defects in anti-inflammatory signaling based on an in vitro study [24]; this was not tested directly in vivo. A similar approach of targeting specific tyrosines engaged in phagocytosis signaling and anti-inflammatory signaling may be employed to dissect the phagocytosis versus anti-inflammatory function of MERTK.

The concept that the efferocytosis and the anti-inflammatory activities of MERTK are indeed dissociable was provided by Tibrewal et al. in 2008 by mutagenesis experiments in vitro [24]. For the first time, a tyrosine residue was identified that governs MERTK-dependent efferocytosis. But this mutation left the anti-inflammatory function of MERTK entirely unaffected. Conversely, the tyrosine responsible for the anti-inflammatory function of MERTK was not identified in this paper, despite extensive mutagenesis. This included mutagenizing the tyrosine within a putative immunoreceptor tyrosine-based inhibitory motif (ITIM). In summary, this provides proof-of-concept that the phagocytosis activity and the anti-inflammatory signaling of this receptor can be dissected. Furthermore, at least the tyrosine-mediating signaling that induces cytoskeletal changes leading to phagocytosis was also identified. The remaining challenge relates to identifying the tyrosine(s) involved in anti-inflammatory signaling. The logical candidate is indeed the tyrosine in the putative ITIM. The existence of ITIM-like sequences in the TAM RTKs was first noted by Lu and Lemke [10], although such sequences are mostly hypothesized to be located in disordered regions [81]. A recent bioinformatic approach employing AlphaFold identifies AXL as a receptor with inhibitory motifs [81]. But targeting the tyrosine in the ITIM of MERTK by Tibrewal et al. failed to disrupt anti-inflammatory signaling. Investigation of the evolutionary homology of MERTK reveals that this putative ITIM within the kinase domain is only found in mammals (Figure 3). This sequence is absent in fish (both cartilaginous and bony). There is a sequence consistent with ITIM in amphibia and reptilia, but it lies outside the kinase domain. This sequence also appears to be absent in birds. Finally, mammalian TYRO3 lacks the putative ITIM (Figure 3). Which tyrosine is likely to be involved in anti-inflammatory signaling? This remains an open question that should be addressed. Sequence gazing indicates that TAM RTKs have a putative immunotyrosine-based switch motif (ITSM; Figure 3). ITSM shares similarities with ITIM. This motif is tyrosine-phosphorylated, akin to ITIMs, by Src family kinases. Following tyrosine phosphorylation, both domains bind SH2-containing adapters. A protein that contains both ITIM and ITSM is PD-1. Importantly, while mutation of the ITIM had little or no effect on PD-1 signaling or activity, mutation of the ITSM abrogated functional T cell exhaustion [82]. Thus, putative ITSM domains in TAM RTKs may be candidates for anti-inflammatory signaling. This theory remains to be tested experimentally.

## 9. Phenotypes in 129 ES Cell-Derived *Mertk*^−/−^ Mice—Exclusively a Reflection of Loss of the MERTK Function or Do They Involve Modifiers?

The 129 ES cell-derived *Mertk*^tm1Gkm/tm1Gkm^ mouse line has been extensively used to describe phenotypes as diverse as microglial function during adult neurogenesis or synapse elimination, Alzheimer’s disease, inflammation in the colon, liver injury, endothelial cell physiology, phagocytosis of apoptotic thymocytes, and cancer [15,16,17,48,83,84,85,86,87,88,89,90,91,92,93,94]. Are all these functions exclusively due to the loss of *Mertk*? Or, much like in the case of the retina, are 129-derived genes involved in the phenotype? Three possibilities come to mind: (i) the phenotype is indeed exclusively due to the loss of *Mertk*; (ii) the phenotype involves the loss of *Mertk* but together with a modifier allele within the 129 region; or (iii) the phenotype is not at all due to the loss of *Mertk* but due to a ‘fellow traveler’ 129 allele. Of the three, of course, the first one is the expected result, and without the studies by Vollrath et al. and Mercau et al., it was taken for granted. The second possibility is exemplified by the retina, wherein *Tyro3* functions as a modifier. The third is the most problematic and may include some cancer models such as YUMM1.7 and GL261. We have evidence that some cancer models are amenable to improvements in anti-tumor immunity when *Mertk* is ablated, and thus, the role of MERTK in anti-tumor immunity is context-specific. This is not surprising given the complexity of tumor subtypes and anti-tumor responses. But solving this might hold the key to improved stratification for MERTK-directed therapies and a better understanding of the biology of MERTK itself.

## 10. Concluding Remarks

Studies led by the Lemke lab over the years, as well as numerous colleagues, have established MERTK and other TAM RTK paralogs as genes with important physiological functions, a number of pathologies associated with altered function, as well as molecules ideal for therapeutic targeting. As noted here, some phenotypes ascribed exclusively to *Mertk*, for example, its role in anti-tumor immunity, may be subject to dissent. A major conclusion from the genetic and molecular studies described here is that one needs to be careful with interpretations of findings made in the 129 ES cell-derived *Mertk*^tm1Gkm/tm1Gkm^ line. This is not a mouse strain with an exclusive loss of *Mertk* but carries a number of gene expression changes due to a region of 129 chromosome 2. Henceforth, using the B6 ES cell-derived *Mertk*^−/−V2^ or *Mertk*^−/−V3^ mouse strain would be superior for the investigation of MERTK function specifically. Given that the *Mertk*^tm1Gkm/tm1Gkm^ line is hypomorphic for *Tyro3*, we also recommend a thorough exploration of the differential function of the TAM paralogs to untangle their individual roles, something that Greg’s lab began to investigate [12,13]. Another critical molecular aspect of TAM function that needs further elaboration is the role of these receptors in phagocytosis versus anti-inflammatory signaling activities. Especially with regards to the role of TAM RTKs in anti-tumor immunity, this might prove consequential. Some of the controversies being generated by these mouse models of TAM RTK knockouts notwithstanding (a word Greg uses frequently in his manuscripts), Greg’s approach to studying TAM biology is universally acclaimed for its beauty, agnostic of the eye of the beholder.

## Figures and Tables

**Figure 1 ijms-25-05299-f001:**
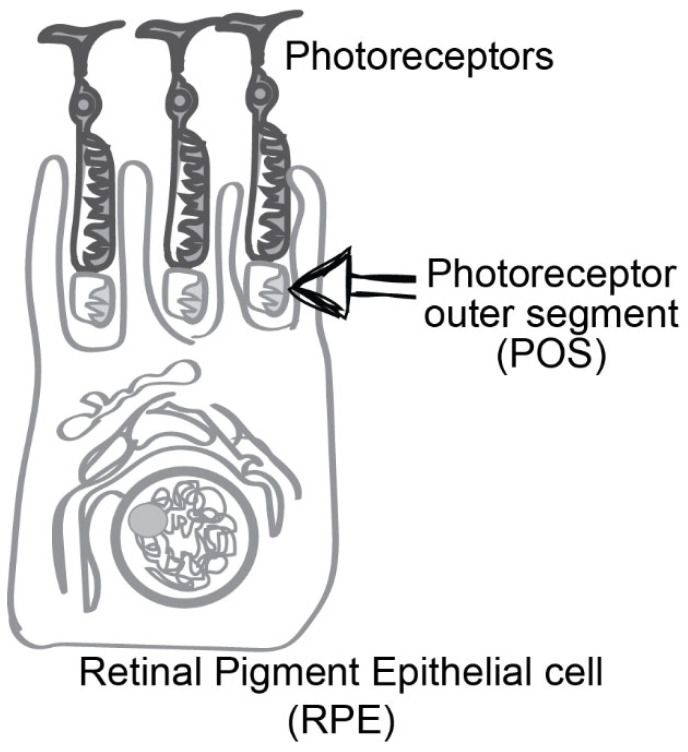
Schematic of photoreceptor outer segment engulfment by a retinal pigment epithelial cell. Photoreceptor cell tips are present in close apposition to retinal pigment epithelial (RPE) cells. The spent photoreceptor outer segments (POS) are engulfed by RPE cells. New discs are renewed in the inner segment of rod photoreceptors, only to subsequently move up and be disposed through RPE-mediated phagocytosis.

**Figure 2 ijms-25-05299-f002:**
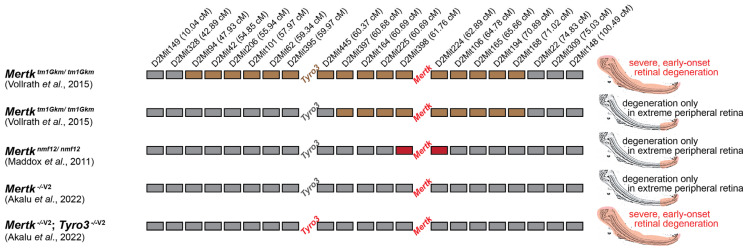
Haplotype maps of mouse chromosome 2 comparing *Mertk* mutant mouse lines and their retinal phenotypes. Comparison of microsatellite markers across chromosome 2 in the indicated mouse lines. Black rectangles indicate homozygosity for C57BL6 alleles, and brown rectangles indicate homozygosity for 129 alleles. Red rectangles indicate the region where the retinal phenotype in *Mertk*^nmf12/nmf12^ mice was mapped to before sequencing the *Mertk* cDNA. The positions of *Tyro3* and *Mertk* are shown. *Tyro3* is indicated in brown when the hypomorphic 129 alleles are present, gray when B6, or red when knocked out. The corresponding retinal phenotype for each line is indicated on the right [42,45,46].

**Figure 3 ijms-25-05299-f003:**
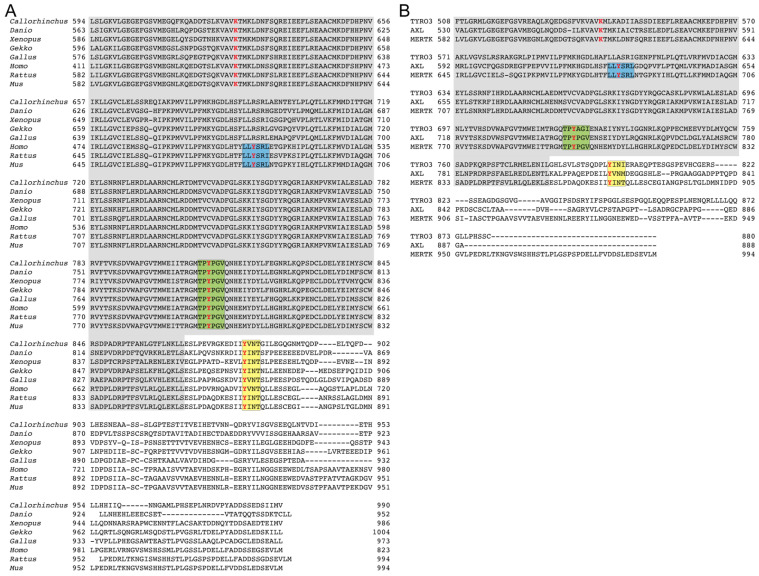
Amino acid sequence alignments identifying conserved domains/motifs in MERTK and its paralogs. (**A**) Amino acid sequence alignment of MERTK kinase (shaded in gray) and kinase extension domains across various genera. The catalytic lysine and critical tyrosines within the putative immunoreceptor tyrosine-containing inhibitory motif (ITIM), immunoreceptor tyrosine-containing switch motif (ITSM), and GRB2/PI3K p85/PLCγ-binding domain are indicated in bold and red letters. ITIM is highlighted in blue, ITSM in green, and the GRB2/PI3K p85/PLCγ-binding motif in yellow. Only motifs within the kinase and kinase extension domains are shown. *Xenopus* and *Gekko* (but not birds) have LIYGVV and ILYLSV sequences at the N-terminal of the kinase domain. (**B**) Amino acid sequence alignment of *Mus* TYRO3, AXL, and MERTK kinase (shaded in gray) and kinase extension domains. The catalytic lysine and critical tyrosines within the putative ITIM, ITSM, and GRB2/PI3K p85/PLCγ-binding domains are indicated in bold and red letters. ITIM is highlighted in blue, ITSM in green, and the GRB2/PI3K p85/PLCγ-binding motif in yellow.

## Data Availability

Not applicable.

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
