# Peer review of "In the Eyes of the Beholder—New Mertk Knockout Mouse and Re-Evaluation of Phagocytosis versus Anti-Inflammatory Functions of MERTK"

_ijms, 2024, doi:10.3390/ijms25105299_

Round 1
Reviewer 1 Report
Comments and Suggestions for Authors
In the present paper “In the eyes of the beholder – new Mertk knockout mouse and re-evaluation of phagocytosis versus anti-inflammatory functions of MERTK”, Sourav Ghosh and coworkers described later studies employing mouse genetics and the generation of newer knockout models that point out incongruencies with the inference that loss of MERTK-dependent phagocytosis is sufficient for severe, early-onset photoreceptor degeneration in mice. Moreover, the authors suggest new molecular approaches that can further Greg Lemke’s scintillating legacy of dissecting the molecular functions of MERTK – a protein that has been described to function in phagocytosis, as well as in negative regulation of inflammation. Overall, I think that the manuscript is intriguing, well-written and well-structured.
I have some small suggestion/curiosity to improve the quality of review.
1) Both MERTK and NF-κB are linked with inflammasome signaling pathways; then, these pathophysiological processes could represent a potential target for new anti-inflammatory compounds. Please add a comment in discussion section of manuscript and please kindly insert appropriate references.
2) MAPK (es. p38, p-ERK) signaling could represent a potential regulation molecular pathway in the present experimental model and, specifically, in MERTK-dependent phagocytosis. Please make a comment about it.
Author Response
We thank the Reviewers for their time, overall positive evaluation, and critical suggestions. We have tried to address the concerns and hope the manuscript is now suitable for acceptance.
Below are our responses to the comments:
Reviewer 1.
“Both MERTK and NF-κB are linked with inflammasome signaling pathways; then, these pathophysiological processes could represent a potential target for new anti-inflammatory compounds. Please add a comment in discussion section of manuscript and please kindly insert appropriate references.”
We have now included a sentence stating:
Of note, MERTK has been directly implicated in the inhibition of NLRP3 inflammasome activation by upregulating autophagy 77.
We also discuss:
The discovery of inflammation as a critical aspect of retinal degeneration associated with loss of MERTK might spur the investigation of anti-inflammatory therapies in alleviating vision loss. Jakinibs 66 or autophagy inhibitors 77 can be used to suppress inflammation resulting from loss of MERTK.
“MAPK (es. p38, p-ERK) signaling could represent a potential regulation molecular pathway in the present experimental model and, specifically, in MERTK-dependent phagocytosis. Please make a comment about it.”
This is an interesting suggestion. MAPK is certainly an important effector pathway downstream the TAM RTKs.
MAPK is also implicated in phagocytosis. However, MAPK pathway is inhibited during TAM RTK anti-inflammatory signaling (Rothlin et al. Cell, 2007). Effect of MERTK on MAPK signaling in RPE is not clearly established. Therefore, we believe that discussing this pathway in the context of phagocytosis would lead to more confusion for the reader.
Reviewer 2 Report
Comments and Suggestions for Authors
This review paper focus on the interaction of Mertk knockout with genetic background and the knowledge gained in terms of the role of Mertk in phagocytosis and anti-inflammatory function focusing mostly on retinal degeneration.
It is well written with only some typos throughout the manuscript. The ideas are mostly clear introduced and presented. I would say that it uses a pretty big number of acronyms and I found myself going back to first occurrences often, so I would suggest adding an abbreviation list somewhere in the paper. I also was a bit puzzled about RCS rat which, I believe, was not defined or at least I missed it and so had to ask my AI assistant!... Along this idea, for someone out of the field, some schematics of POS, RPE and eye-related biology talked about here would be useful. I found the historical description of the different mouse models and their different genetic backgrounds a bit long and I believe there should be way to shorten this a bit and focus a bit more on some more recent discoveries or maybe explain a bit better the biology that the paper is focusing on here.
Section 4 lines 163 to 196… I felt that the importance of the genetic background difference was not so convincing given that MERTK-K614M was having a different mutation… or I may have missed the point, in which case, I guess the text needs some clarification.
Comments on the Quality of English LanguageVery good just a few minor typos.
Author Response
We thank the Reviewers for their time, overall positive evaluation, and critical suggestions. We have tried to address the concerns and hope the manuscript is now suitable for acceptance.
Below are our responses to the comments:
Reviewer 2.
“…I would suggest adding an abbreviation list….”
We have added a List of abbreviation section.
“…RCS rat which, I believe, was not defined..”
We apologize for this omission during edits. We have now stated:
The molecular understanding of MERTK mutations in human RP was preceded by observations of a spontaneous animal model now known as Royal College of Surgeons (RCS) rat.
“…some schematics of POS, RPE and eye-related biology talked about here would be useful”
We have included a schematic (new Figure 1).
“I found the historical description of the different mouse models and their different genetic backgrounds a bit long and I believe there should be way to shorten this a bit and focus a bit more on some more recent discoveries or maybe explain a bit better the biology that the paper is focusing on here.”
We feel the historical description is required to provide the context for the findings made with the 129 embryonic cell-derived Mertk -/- mouse line and the later discoveries with B6 Mertk -/- mice. But, following the suggestions of the Reviewer we now try to explain the biology by stating:
In summary, these studies identify that severe, early-onset retinal degeneration and vision loss upon loss of Mertk in mice was dependent on genetic background and not solely due to the absence of the Mertk gene.
“Section 4 lines 163 to 196… I felt that the importance of the genetic background difference was not so convincing given that MERTK-K614M was having a different mutation… or I may have missed the point, in which case, I guess the text needs some clarification.”
We re-wrote the section as follows:
The authors considered allelic differences as an alternate possibility, Mertk nmf12 being a missense mutation as opposed to the presumptive null allele that is Mertk tm1Gkm. A 2.5-fold reduction in MERTK protein in Mertk nmf12/nmf12 mice compared to WT B6 mice was demonstrated in the publication. Critical residues essential for receptor tyrosine kinases activity includes HRD motif 43 (the phenylalanine of the DFG motif makes hydrophobic contacts with the HRD motif) as well as the ATP coordinating lysine. Given that the mutation in Mertk nmf12 mapped to the HRD motif, this mutation would almost certainly render the Mertk nmf12 derived protein kinase inactive. Thus, in hindsight, the phenotypic difference was due to differences in genetic backgrounds and not due to the missense mutation retaining significant level of phagocytic activity. Work in the Rothlin-Ghosh laboratory later demonstrated that in fact kinase-inactive MERTK generated by substituting the ATP-coordinating lysine in B6 mice (Mertk K614M/K614M) had a similar reduction in macrophage phagocytosis of apoptotic cells as macrophages from Mertk tm1Gkm/tm1Gkm mice 44.